# Guidelines in the Preparation of Fully Synthetic, Human Single-Domain Antibody Phage Display Libraries

**DOI:** 10.3390/antib14030071

**Published:** 2025-08-15

**Authors:** Mark A. Tornetta, Brian P. Whitaker, Olivia M. Cantwell, Peter N. Haytko, Eileen D. Pisors, Fulai Zhou, Mark L. Chiu

**Affiliations:** 1Tavotek Biotherapeutics, 727 Norristown Road, Spring House Innovation Park, Building 3, Suite 101, Lower Gwynedd, PA 19002, USA; brian.whitaker@tavotek.com (B.P.W.); olivia.cantwell@tavotek.com (O.M.C.); peter.haytko@tavotek.com (P.N.H.); eileen.pisors@tavotek.com (E.D.P.); mark.chiu@tavotek.com (M.L.C.); 2Tavotek Biotherapeutics Building C2, Suzhou Biomedical Industrial Park, Suzhou 215000, China; fulai.zhou@tavotek.com

**Keywords:** phage display, synthetic, nanobody, single-domain antibody, variable heavy chain

## Abstract

Background/Objectives: The complexity of diseases such as cancer and auto-immune disorders drives the need for unique, target-driven therapeutics. A broader arsenal to generate better biologics-based therapeutics is needed to provide more efficient and effective antibody generation technologies. The critical parameter for antibody generation is to generate as much candidate diversity to each target as possible. Method/Results: We present guidelines for having an efficient process using a fully synthetic human single-domain antibody (sdAb) phage display library. Critical milestones for success focused on library quality control (QC) assessments, evaluation of specific biopanning outputs, and construct designs that enabled efficient transition to mammalian expression. The synthetic VHO libraries produced epitope diversity better than an immunized sourced library with candidates possessing nM potencies and monodispersity > 90% via SEC. Conclusions: Synthetic human scaffold sdAb phage display libraries was constructed, biopanned, and selected candidates that could be directly transitioned for mammalian expression. The diverse VHO sets of candidates produced from many targets easily provided opportunities to make a multi-specific biological compound. Both synthetic and immunized phage selection campaign results suggested that these technologies complemented each other to generate therapeutic candidates. Finally, we demonstrated how diverse data produced from a process that used VHO synthetic libraries could accelerate drug discovery.

## 1. Introduction

Rising demands to have specific protein-based binders for applications in pharmaceuticals, diagnostics, synthetic materials, and nanotechnology require more effective methods to select a range of binding variants. Typically, diverse protein binders are selected from oligonucleotide-, peptide-, protein-, and oligosaccharide-based libraries. However, there is an art of how best to screen such libraries to identify the desired binders. Although the use of genetic methodologies and display technologies have improved the speed and reduced the resources needed to generate biologic candidates, there still is a need for more efficient methods to obtain lead molecules [1,2,3,4,5].

Over the past few decades, hybridoma and display technologies have been integrated into high-throughput industry settings to generate antibody-based molecules [6]. While the conventional selection goal is to generate a candidate with the most potent efficacy, we are learning that these strategies have produced candidates that have not reached the desired efficacy in clinical development [7]. The main challenges remain in increasing the candidate’s specificity to the disease target as well as in having more diverse candidates that have the relevant mechanisms of action for the lead optimization stage in the discovery process. Thus, a successful readout of a library is the generation of a diverse candidate pool for the target. When we say diverse, we are referring to many epitopes represented within that candidate pool that still have protein stability for developability.

A broader candidate pool first requires having diverse collections of DNA sequences presented in a stable centralized carrier format using a peptide, antibody, or protein binder framework [8]. The gene library repertoires can come from natural (either naïve or un-immunized and immunized animals) or synthetic sources (semi or fully synthetic) (Figure 1). Typically, careful applications of error-free methods to capture DNA sequences using direct cloning, PCR, and gene synthesis into DNA plasmid or viral phagemid vectors are critical for fidelity in the recombinant DNA manipulations [9,10].

Historically, animal-sourced libraries have produced high affinity, manufacturable antibodies that have transitioned into clinical trials and regulatory approvals [11,12,13]. However, immunization-based antibody response can be constrained by tolerance and ‘self’ recognition, which results in the animal response bias to the antigen resulting in poor diversity and the lack of species cross-reactivity [10]. In addition, a faulty PCR process lacking sufficient oligonucleotide codon coverage can result in lower diversity from an immunized repertoire [14]. Also, the conventional in vitro high-throughput screening methods that select candidates based on more potent binding affinities have produced candidates with less diverse therapeutic activities [11,12,13,15].

Synthetic library selection and biopanning can create more diverse candidates than that of human and animal immune responses to antigenic challenges. It is also worth noting that candidates from synthetic libraries, compared to those from an animal immunization, do not have to undergo humanization process, which is both time- and cost-consuming [16]. Rather, the synthetic approach requires efficient selection of more diverse candidate pools. Typically, the resulting candidates can have lower potencies in the hit-to-lead selection. In addition, synthetic antibody candidate sequences often need to be optimized to have favorable manufacturability or developability profiles [17].

Presently, there is a strong demand for single-domain antibody (sdAb) or nanobody applications in antibody-based engineering. The sdAb size is smaller, enabling the generation of more paratopes to interact with a greater range of epitopes on the target. While Fabs and scFvs with larger contact footprints can possess better affinities, their size can limit the diversity of epitope engagement via steric hinderance. However, the sdAbs can be engineered to be a tandem sdAb in multispecific molecule format to improve upon disease target specificity. In addition, the sdAb can have lower pH selectivity and higher temperature stabilities compared to Fab and scFv binders. In addition, Fab and scFv domains require interchain association of both domains/chains for stability and activity [18,19].

Establishing a sdAb library requires having an antibody(s) format that has expression stability in the host used for selection. While some antibody frameworks are not stable for bacteriophage, yeast, or mammalian display [1,20,21,22], other synthetic phage display libraries are employed, but with a limited number of frameworks [18,19]. Each display technology has unique challenges: yeast display suffers from resulting in proteins with non-human post translational states [20], mammalian display can have inconsistent expression, and alternative display technology can require more resourcing for best performance [14,15,16,17]. Although antibody expression can be instable in *E. coli*, bacteriophage (phage) display technology has improved to express and display enough types of antibody fragments, thereby making the process more versatile for selections/biopanning [21,22,23,24].

We have compared sdAb technologies from the public domain to gain further insight into certain unmet needs of sdAb libraries in Table 1 [24]. Such libraries have been established from immunizations, semi-synthetic (combination of immunized and synthetic regions), and fully synthetic libraries. Unfortunately, only a few of the sources in Table 1 describe a high percentage of good developability behaviors. Good developability produced from a sdAb generation campaign includes the following: the level of hit diversity as shown by the number of paratopes, or CDR3 sequence, as well as the number of activity bins or epitope coverage. Most of the sdAb phage libraries have been sourced from ‘natural’ antigen-immunized libraries. To improve upon the advantages of having a fully synthetic sdAb phage library, several considerations are required. Here, we demonstrate a process to overcome the challenges of DNA synthesis, construct cloning, expression, and display of a fully synthetic sdAb library. We present how an M13 filamentous bacteriophage display technology and a type of sdAb, VHO (variable heavy only), library could allow for all three CDRs to be diversified within a fully human variable heavy chain framework. While we show how this technology was an efficient process for hit generation [25], we also present our experience for how to obtain a successful fully human synthetic human sdAb library that can continuously generate diverse candidates.

**Figure 1 antibodies-14-00071-f001:**
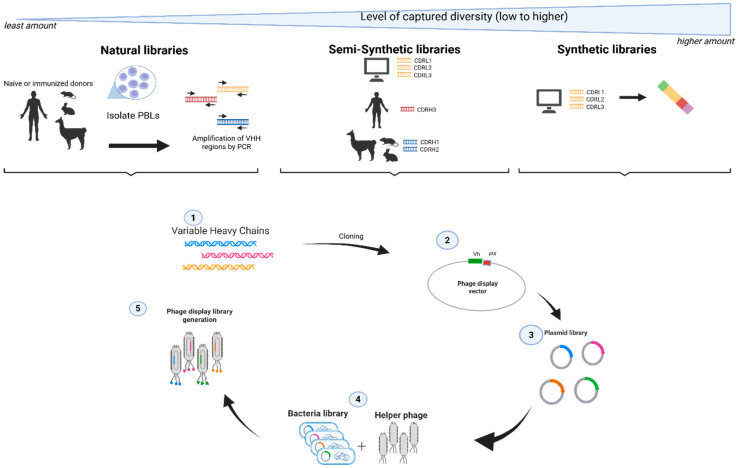
Profile of different antibody phage libraries [13]. Antibody phage libraries can be sourced from natural libraries, semi-synthetic libraries, or fully synthetic libraries. Diversity is defined by the number of different variable region sequences. CDRH and CDRL are complementary determining regions of the heavy chain and light chain, respectively. Arrows along with numbering show the stepwise direction of a library construction process.

## 2. Materials and Methods

Bacterial, phage, and tissue culture products were obtained from Teknova (Hollister, CA, USA), KD Medical (Columbia, MD, USA), Thermo Fisher Scientific (Waltham, MA, USA), Invitrogen (Carlsbad, CA, USA), Gibco (Grand Island, NY, USA), and Sigma (Burlington, MA, USA). Cell lines were obtained from ATCC (Manassas, VA, USA), Lucigen (Middleton, MI, USA), and VCSM13 phage from Agilent Technologies (Santa Clara, CA, USA). Telesis Bio (San Diego, CA, USA), Gene Art (Regensburg, Germany), Integrated DNA Technologies (Coralville, IA, USA), Genewiz (South Plainfield, NJ, USA), and GenScript (Piscataway, NJ, USA) performed gene synthesis for the phage library and expression constructs. Target proteins were acquired from ACRO Biosystems (Newark, DE, USA), Sino Biological (Wayne, PA, USA), and R&D Systems of Bio-Techne (Minneapolis, MN, USA).

### 2.1. VHO Scaffold Phage Display Libraries

A human IGHV3 consensus from some of the sub-families (IMGT) was used as the scaffold to build the libraries [50]. The diversity of amino acid combinations within CDR1 and CDR2 were limited to less than 10^5^ excluding C [51,52]. The diversity of amino acid combinations for CDR3 is greater than 10^14^. Each library design consisted of 8 different lengths of CDR3 either minimizing or excluding C, M, and N. The first design used a trinucleotide-based technology [53] to synthesize and assemble the mutational fragments (full control of codon use). The second design was based on traditional oligo synthesis and Gibson assembly performed by Codex/Telesis Bio (reduced control of codon use). Amino acid is referenced by a single letter code. The CDR definitions were based on Kabat classification [54]. Library construction into a dual-functional phagemid, transformed, propagated, and infected for the final VHO phage display libraries was described by Tornetta et al. [25]. Final phage libraries were subjected to NGS sequencing to evaluate the diversity and functionality of the libraries. In addition, portions of libraries were frozen pre-infection as glycerol stocks and used later to infect an additional lot of the same constructed libraries.

### 2.2. VHO Phage Biopanning to Generate Target Specific Candidates

A KingFisher Apex system from Thermo Fisher Scientific was used for high-throughput phage biopanning. The capture of biotinylated or polyhistidine-tagged target proteins was performed by using Streptavidin-coated magnetic beads (Invitrogen™, Carlsbad, CA, USA; Dynabeads™ M-280 Streptavidin, Thermo Fisher Scientific, Waltham, MA, USA) and/or nickel-coated beads (Thermo Fisher TM Ni-NTA magnetic beads), respectively. The KingFisher magnet was used to collect the target-coated magnetic beads after the phage library was incubated with the target protein-coated beads. The unbound phage was removed in subsequent wash steps. Target/bead bound phages were captured by incubating/infecting with log-phase MC1061 F’ cells. Additional biopanning rounds were performed against decreasing concentrations of the target protein. At least one round of biopanning was performed using only the magnetic beads without the target antigen present to eliminate non-specific VHO phages. To maximize VHO candidate diversity, the library was presented to human, mouse, and cynomolgus protein orthologs, full length as well as extracellular domains, expressed in mammalian cells displaying the target via cell panning methods. The biopanning was also conducted under an acidic pH of 6.0 commonly found in disease environments. After the last round of selection, phagemid DNA was obtained to be used for NGS.

### 2.3. Next Generation Sequencing and VHO Sequence Processing

The final round of extracted phagemid DNA was used to create NGS library amplicons capturing all three CDR sequences with primers located in Framework 1 and Framework 4 using the Telesis Bio BioXp™ model 3200 equipped with their NGS amplicon library program. The MiSeq NGS reading technology (Illumina, San Diego, CA, USA) was used to generate up to 25 million sequences, while the Nextgen NGS technology (Illumina, San Diego, CA, USA) was used to generate more than 50 million sequences. The handling of the large amount of sequence data was performed by PipeBio software (PipeBio/Benchling) which performed most of the processing that included merging paired-end sequence reads, aligning and translating the VHO amino acid framework, counting the enrichment of all the clones reads, identifying and grouping any CDR3 sequences with >90% similarity, and merging data between multiple project biopanning groups.

### 2.4. VHO-Fc Expression Plasmid Construction via Automated Cloning

From the NGS processing software (PipeBio), VHO sequences were used to create customized DNA cloning kits from Telesis Bio. The BioXp™ platform (TelesisBio v1.3.53) was used to perform VHO fragment assembly and ligation into a pcDNA3.4-based mammalian expression plasmid (Thermo Fisher). All VHO sequences were fused from the c-terminal end to a G4S linker which was fused upstream of the hinge (including cysteines) of a human IgG1 Fc (VHO-Fc). All transformed, single-colony grown DNA preparations were sequenced by Sanger (Azenta Life Sciences, Burlington, MA, USA or Eton Bioscience Inc., San Diego, CA, USA) [25].

### 2.5. Transfections and Automated VHO-Fc Purifications

An Expi293 transient expression kit from Gibco^TM^ was used to transfect and transiently express each VHO-Fc candidate. The harvested media supernatants after 5 days of incubation were then processed using the KingFisher automated system (Thermo Fisher) and protein A or G magnetic beads (Promega, Madison, WI, USA). The purified VHO-Fc molecules were analyzed by SDS-PAGE as well as by SEC [25].

### 2.6. Biolayer Interferometry Kinetics

Binding kinetics were performed on a Gator BLI instrument following vendor-recommended standard operating protocols, as described by [42]. The human, mouse, and cynomolgus monkey target protein variants were captured to Ni-NTA sensor probes (Gator Bio, Palo Alto, CA, USA) if HIS tagged, and captured to streptavidin sensor probes if the target proteins were biotinylated [25]. BLI binding activity was scored according to the response signal of >0.2 nm at the point of equilibrium (no further change in BLI response) during the analyte association phase. To establish specificity and/or background, both a negative target protein and a negative VHO-Fc were used.

### 2.7. ELISA Binding

The target protein was attached to Maxisorp plates (Nunc, Carlsbad, CA, USA), either directly immobilized to the plate surface or through streptavidin interaction with the biotinylated target [25]. Negative controls consisted of VHO molecules specific to unrelated target proteins and positive controls were well-characterized, target specific IgG molecules. The positive binding profile was determined as a signal exceeding three times that of the negative controls at a concentration of 10 nM or less.

## 3. Results

### 3.1. Library Design

We used the TavoSelect phage display platform to generate the VHO sdAb [25]. The VHO library used a consensus of some human VH3 frameworks (IMGT) to capture diversities in all three CDRs and had a theoretical combined diversity of 1 × 10^5^ for CDR1 and CDR2. With two different designs and each consisting of eight different lengths of CDR3, the range of theoretical diversity was between 1 × 10^11^ to 1 × 10^19^. There were several criteria to establish a working VHO library. One of the most important criteria was having the library quality control (QC) of >10^8^ transformation efficiency per ligation into a phagemid, and the ORF from CDR1 through CDR3 to be >40%. Another key criterion was to establish a high level of confidence that the 5′ end (FW1) and the 3′ end (FW4) were >95% accurate via Sanger sequencing during the DNA synthesis assembly process.

### 3.2. Library QC

The whole VHO library, comprising 16 VHO sub-libraries with transformation efficiencies ranging from 5 × 10^8^–2 × 10^9^ colonies, created a total diversity of 2 × 10^10^. Two lots of each library were created, with lot 002 being made 3 years later than lot 001. NGS analysis of the sub-libraries was compared using two different NGS technologies. The NGS technology for lot 1 using MiSeq (Illumina) produced 5–5.5 million sequences and lot 002 using NextSeq (Illumina) produced 34–40 million sequences. The level of the VHO design 1 library functionality, defined as being in the reading frame from CDR1 through CDR3, had no extra C, and followed the designed CDR3 length, was 69% for lot 001 and 64% for lot 002 (Table 2). Design 2 library functionality was 36% and 35%, respectively, for each lot (Table 2). Lot 1 provided years and hundreds of biopannings in generating VHO candidates.

Table 2 showed the quality control results on the original lot 1 library construction and on the lot 2 library construction. The library NGS QC profiles of the two designs demonstrated comparability.

### 3.3. Biopanning Output (NGS Sequence)

The different presentations of the targets by way of using animal orthologs, having various fusion tags, and in pH 6 or pH 7 environments were applied to guide the NGS processing and analysis of the selected VHOs after biopanning. This revealed many unique CDR3 sequences, as shown in a LOGO plot of 1849 VHO candidates generated after 25 biopanning campaigns (Figure 2A). The CDR3 length distribution of these >1800 VHO phage display biopanning outputs showed a bell-shape distribution (Figure 2B) like that which has been published in the literature regarding human IgG heavy chain CDR3 lengths [24].

The risk-assessed VHO sequences generated from 25 biopanning campaigns showed very little difference in hydrophobicity (Geneious Prime software version 2025.1, built on 19 May 2025) compared to the positive primary screened VHH sequences from 3 llama immunization campaigns. Accounting for the whole variable region sequence, the average hydrophobicity for 1849 VHOs was 47.6%, whereas the average for the 182 VHHs was 47.0%. Including the sequence region from CDR1 through CDR3 (not including FW1 and FW4), the hydrophobicity averages for VHO were 47.2% and for VHH was 45.9%. For the CDR3 region, the hydrophobicity average for the VHOs was 45.1% and for the VHHs was 41.3%.

The developability risks within the CDR3 of the 1849 target selected VHOs were 5 N-linked glycosylation sites, 20 potential deamidation sites, and 259 potential isomerization sites. Most of these PTM risks are due to library design 2, the reduced codon control method. As for developability risks found in the CDR3 of the 182 VHH primary screened hits from the immunization campaigns, there were 9 VHHs with C residues in both CDR2 and CDR3 creating a second intra-disulfide bridge, 9 potential deamidation sites, and 33 potential isomerization sites. The PTM risk rate within CDR3 for the VHOs was 15% and for the VHHs it was 28%.

### 3.4. Mammalian Expression of the Candidates

The VHO-Fc candidates were expressed in a 3 mL transient Expi293 system (Thermo Fisher) followed by a Magne Protein G beads (Promega) purification procedure using the KingFisher (Thermo Fisher), which yielded a range of 80–600 µg of protein, whereas 68% of the VHO–Fc proteins yielded >400 µ g. The monodispersity assessment of the VHO-Fc candidates showed approximately half of them having a main peak, measured at an OD280 in units of nm, greater than 80% of the total protein assessed by SEC (GE Biosciences, Chicago, IL, USA). See Tornetta et al., 2024 [25] for examples of what SEC profiles as well as SDS-PAGE gel images of VHO-Fc proteins look like.

### 3.5. Candidate Activity Results

The target protein binding results of VHO-Fc candidates from 12 tumor-associated antigen projects are shown in Table 3. At least 75% of the VHO-Fc proteins possessed binding activity to the recombinant human TAA protein format used during the biopanning. The binding affinities to human TAA protein had a median range KD of 1–10 nM [25]. Every project was successful in generating a cross-reactive VHO-Fc that was either biopanned and/or screened against an ortholog TAA protein. Table 3 shows the average percentage of VHO-Fcs cross-reactive to mouse/rat at 76% and cross-reactive to monkey at 79% [25]. On average there are ~10 epitope bins from 12 TAA projects. The different binding patterns were established from different presentations of the TAA protein (i.e., immobilized TAA by ELISA vs. affinity tag displayed TAA by BLI), ortholog TAAs (cross-reactive activity patterns), cell surface expressed TAA binding, or by competitive binding assay between the VHO candidates [25].

## 4. Discussion

We presented guidelines for a highly efficient platform consisting of an effective and diverse library going into the hit generation stage (biopanning), which ultimately leads to efficient hit-to-lead characterization assessment. The resulting multiple candidates per target could be another means to address how to develop more successful biological candidates for the clinic. We highlighted how the synthetic sdAb platform generated more diverse candidates with good protein behavior and single nM potency. The sdAb scaffold have been used to build multi-specifics that can add value to any biological moiety.

From previous experiences in biologic discovery [56], the decision to use sdAbs was due to its intrinsically favorable structural and biophysical properties [57]. Another beneficial sdAb feature of having smaller paratopes can enable more epitope coverage per target. To take full advantage of these properties, a highly diverse sdAb generation technology was established by implementing at least four of the following parameters: framework, level of CDR diversity, display technology, and an efficient process to select the most diverse hits.

Our strategy of building an effective diverse library involved creating a fully synthetic human framework sdAb, called VHO. From our experiences with synthetic human Fab phage display technologies [52] and the VH3, VH4, and VH1 germline sub-families, we built a library using a heavy variable region framework. A consensus of human IGHV3 (Figure 3, Human_VH3) was used because of its success in other synthetic phage display libraries and was found to have good biophysical attributes as well as reduced immunogenicity [27,58,59,60]. The strategy to design a very diverse library was also intended to overcome the challenges with using only a human variable heavy domain [36].

Diversity was established by applying degenerate mutagenesis methods to all three CDRs, varying the CDR3 length, and applying two different CDR3 mutagenesis methods. Keeping in mind that antibody variable domains can be GC rich and possess levels of consensus sequence stretches, which can impact the efficiency of DNA synthesis, we performed more than one methodology. We found that trinucleotide-based oligos that provided specific residue substitutions were dependent on DNA sequence annealing, while traditional oligo synthesis (NNK, NNM, etc.) provided less challenges for annealing. Take note that using a fixed codon use for expression will affect DNA assembly during the library construction process. We observed that trinucleotide-based synthesis provided an almost perfect expected design. The more traditional NNM/NNK oligo synthesis, as expected, could not keep all undesirable codons from being synthesized [61] while minimizing impacts on DNA annealing due to novel codon sequences. In our hands, Gibson assembly provided better assembly accuracy than other annealing techniques but the method limited the number of juxtaposed residues for diversification.

To avoid creating a non- or less-functional synthetic library due to inappropriate or excessive changes per variable region, we took a balanced approach to sequence diversity. We first limited the sequence diversity within CDR1 and CDR2 to changes that were previously observed in IGHV3-based antibodies [25]. A liberal use of degenerate codons within CDR3 imbued the library with a highly diverse repertory. The success of our balanced mutagenesis approach was demonstrated by the diverse sets of candidates to a variety of targets (Table 3). We continued to learn from the diverse sets of VHOs and the relationships between the residues that made up the frameworks and the level of diversity in the CDRs. The overall designing of such libraries integrated DNA synthesis, expression, and display on phage to create functional proteins.

Other design factors included the consideration that the llama, camel, and alpaca VHHs with longer CDR3 regions than that of humans could translate to unattainably large library sizes. By focusing on the CDR3 loop length distribution found in humans [62], we utilized the eight predominant lengths with the longest CDR3 length to create seventeen residues (following the KABAT residue format). Additional logistical parameters included working with 10^8^ log phase host bacteria cells per mL, 10 transformations averaging ~10^9^ transformation efficiency per CDR3 length library, and 2 sets of library designs, which all collectively generated a 2 × 10^10^ size library. We found that this first lot of the library covered hundreds of biopannings over a period of 4 years.

As far as the VHO gene and phagemid assembly process, we found traditional restriction enzymes and T4 ligase gave the best transformation results compared to other recombinant DNA splicing techniques. A phagemid has the appropriate elements for DNA replication for both bacteria and phage, as well as effective antibiotic resistance. The expression cassette within the phagemid contained a promoter, a signal sequence, a multiple cloning site with efficient restriction enzyme sites, and the surface phage coat protein as the fusion partner that displayed the library of VHOs on the surface of the bacteriophage [63,64]. We chose to fuse the VHO on pIX because it was not involved in F pili fusion like that of pIII [52]. Nonetheless with enough resources, we would also display the VHO library to pIII to take advantage of its longer length to provide more flexibility for the VHOs to interact with the target during biopanning.

We found that using a particular host bacterial cell type (TG1, XL-1, MC1061) for certain steps of the library construction process yielded better transformation efficiency, phage library amplification, expansion, and propagation of VHO containing phages. The type of transformation method, such as heat versus electroporation, was dependent on the host cell competency (chemical treatment and the density of cells). The amount of DNA added to competent cells was also critical. Too much ligated DNA produced a lower transformation efficiency, just like using too little. In addition, determining when and how much pooling occurred was important for the growth process after transformation. After the recovery growth period, a sample of the culture was assessed for transformation efficiency. The resulting process affected the amount of the designed library size that was captured in the bacterial host.

Performing as much quality control assessment on the libraries before investing resources, such as biopanning, was imperative for success. At the VHO phage library stage, NGS and sequence analysis were used to evaluate the percentage of in-frame sequence (% functionality) and realistic diversity (looking for duplicate sequences) within the libraries. The % functional differences between the two designs were expected since design 1 was performed by trinucleotide oligo technology (full control in codon use) and design 2 was performed by a degenerate oligo mixing method (reduced control in codon use). Nonetheless, the functionality percentages between the lots were comparable. The trimer oligo technology was reported to be >80% functional libraries [23,24], and this was lower than expected. An explanation for the lower-than-expected functionality was due to the methods used for fragment assembling during the synthesis process. Oligo technologies played a very important role for the annealing and assembly steps of a synthesis project [23,24].

Applying more than one biopanning parameter was also crucial to capturing as much candidate diversity to a given target. For example, we implemented the use of animal orthologs of each target protein not to just produce cross-reactive VHOs [25] but to increase epitope diversity within the candidate pool too. Biopanning was also performed at both pH 6 and pH 7 to provide higher levels of selectivity to the disease environment (cancer or inflammation) from normal tissue environment conditions, respectively [25,63]. Due to such a diverse fashion of biopanning performed, a dependence on NGS was the only way to capture almost every selected VHO. As expected, we observed from the NGS processing and analyses that many unique CDR3 sequences were captured. As shown in Figure 2A, the CDR3 residue diversity indicated the vast library’s design and revealed the diverse biopanning methods used. The limits on the synthesis of library design 2 are represented in Figure 2A by the over representation of residue G. In short, this was a side effect of using traditional NNK/NNM degenerate oligos and Gibson methods during library synthesis and assembly [65].

We selected VHO candidates with minimal developability risks [65] and their unique sequences defined by the diverse biopanning methods used. Each VHO was constructed as an Fc fusion for transient expression in HEK293 cells. We believed expression in mammalian cells for activity characterization was a key attribute since the ultimate use for VHOs was to make highly specific-based biologic compounds. Bacterial expressions, although robust, did limit the size of the expressed product. While yeast expression could handle the size of a multi-specific biologic, different post translational modifications, such as glycosylation, could cause issues in development and in the clinic [22].

The fusion of a VHO to an IgG1 Fc domain could increase the half-life and stability of such a non-traditional protein antibody [66]. The Fc fusion was also an effective backbone during the hit-to-lead screening as well as lead optimization assessment, especially for the selection of multi-specifics. Implementing VHOs as Fc fusions also provided a variety of ways to screen the diverse sets of candidates in characterization assays, such as providing additional size for increased signals with BLI and secondary reagents for enhanced detection with flow cytometry. Routinely, between 48 and 96 VHO candidates of each target gave rise to a plethora of paratopes which resulted in many epitopes or activities bins (Table 3).

From our earliest target campaigns, we noticed the challenges that come with sdAbs. Despite our best efforts in the design of the library, biopanning strategies, and selecting candidates with the least PTMs, we still ended up with challenges during the characterization steps. Having a smaller sized paratope interaction and not undergoing a maturation process led to molecules with lower target affinities that could reduce inhibition activity, when compared to larger paratope or immunized sourced scFvs and Fabs. We addressed the weaker affinity of the first round of VHOs with a 5–10-fold affinity improvement by performing a maturation process that re-diversified the CDRs followed by a stringent biopanning procedure. Another way to improve target interaction could be by using multiple VHOs to create an avidity effect.

Since the synthetic VHOs had a common human HC framework, a consensus of some IGHV3 sub-families, which were normally paired with an LC, meant there could be higher levels of aggregated species; exposed residues that were normally found in the association of the two variable regions. The VHO developability challenges could be circumvented via a combination of framework and CDR mutations that could be implemented either in the de novo library or in an optimization process [27]. We are currently implementing another approach by adjusting the formulation for elution and neutralization steps during affinity chromatography steps. Nonetheless, there were enough VHO hits from biopanning the synthetic library to identify well-behaved candidates.

Referenced in Bahara et al., 2016 [26] they used human VH3–23 as a scaffold and performed randomized mutagenesis to all three CDRs. However, their sequenced library QC results showed unnatural use of certain residues within the CDR regions. After selections against *Mycobacterium tuberculosis* a-crystallin, all but two hits contained stop codons and/or cysteine residues which limited the number of active candidates. The other two candidates contained prolines and methionines within the CDRs where both could contribute to developability challenges. The unnatural use of residues, especially in CDR1 and CDR2, and the frequency of the stop and cysteine codons could be improved by library design and construction methodologies mentioned previously.

Based on previous experiences from Fab-based synthetic and immunized-antibody approaches and how both technologies produced additional candidates, both the de novo synthetic VHO phage display and immunized VHH phage library candidate generation procedures were interacted to the same target proteins. After biopanning, the NGS sequence results for each of the two technology libraries against the same three different targets revealed differences between them. Figure 4 showed an alignment of the most diverse residue arrangement as well as the most surface exposed region, CDR3 [67], for each project per each technology. This CDR3-defined paratope of the VHOs and VHHs was a reflection on antigen epitope recognition. The synthetic VHO candidates showed more diversity in the CDR3 regions compared to the immunized VHH candidates. The immunized CDR3 diversities with these three targets were 59%, 44%, and 70%; while the synthetic CDR3 diversity with the same targets were 87%, 89%, and 94% (Figure 4, respectively, calculated from left to right). Continuing with the comparisons between synthetic and immunized sdAbs were protein behavior (monodispersity) and potency (binding activity). As expected, we found almost all the VHH candidates have good developability behavior compared to 50–80% good behavior within the VHO candidates for the three target protein selections. Also expected was the average of the synthetic VHOs having up to a 10-fold worst potency to targets compared to immunized VHHs. Both important traits were most likely because of somatic mutations that occurred during immunizations. We found residue diversity across the frameworks of the VHHs most likely a relevant result of better developability and potency traits regarding llama/camel immunizations. Contrary to these better behaviors was that such framework diversity could increase immunogenicity. The pros and cons between immunized/natural and phage display/synthetic antibody libraries now have been observed by us in both Fab- and sdAb-based technologies. The pros for immunized libraries include having good developability and potent affinity; and for synthetic libraries having diverse target coverage and less resource requirements. Thus, both natural- and synthetic-sourced antibody technologies could be complementary for candidate generation [27].

Much can be learned from sdAb generation technology. This ‘simpler’ antibody format could continue to provide further understanding of how each paratope played a role in developability and target recognition. The synthetic libraries that successfully produced vast arrays of candidates possessing different biophysical behaviors as well as target activities could enable AI/ML to develop better antibody engineering. Directly linking the candidate sequences to the protein behavior and target engagement activities could move us toward the day when we could predict the very best candidates right from biopanning or immunizations.

## 5. Conclusions

Discussed here were guidelines in constructing and evaluating a synthetic human variable heavy framework, with diversification in all three CDRs of a sdAb or VHO library. Presented here was a profile of a successful library with >99.9% diversity and with a transformation efficiency of 2 × 10^10^ which could generate from a biopanning campaign on average ~10 epitopes per target, possessing an average potency range of 1–10 nM. While >50% of the VHO proteins were monodisperse, there were still enough developable hits to select for the next steps that led to therapeutic candidates [25]. Continued development of synthetic libraries from Fab to now sdAbshowed how the bacteriophage platform evolved into a more efficient biologics discovery process. Continued observation from NGS analyses after biopanning or screening of candidates from either a synthetic or immunized library could be complementary in the outcome of generating successful leads. By implementing a unique human synthetic sdAb, we have shown how our platform increased the ability to efficiently generate a diverse array of candidates to create differentiated antibody-based biologics.

## Figures and Tables

**Figure 2 antibodies-14-00071-f002:**
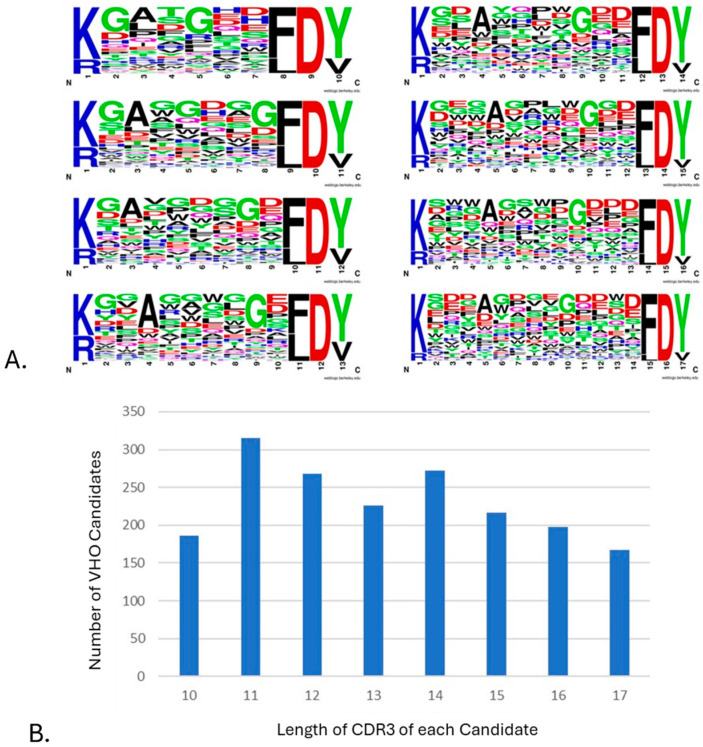
A view of the diversity within 1849 VHO candidates generated from 25 biopanning campaigns. (**A**) High diversity of CDR3 sequences seen after biopanning and NGS candidate selection for each length 10 through 17 [55]. (**B**) CDR3 length observed from VHO candidates after biopanning and NGS analysis.

**Figure 3 antibodies-14-00071-f003:**
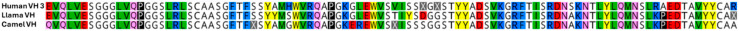
Alignment of the consensus VH sequences from human, llama, and camel (IMGT). The colors are from the Clustal alignment rules. Red for negative charge, blue for positive charge, yellow for aromatic, green for hydrophobic, and pink for polar uncharged amino acid residues.

**Figure 4 antibodies-14-00071-f004:**
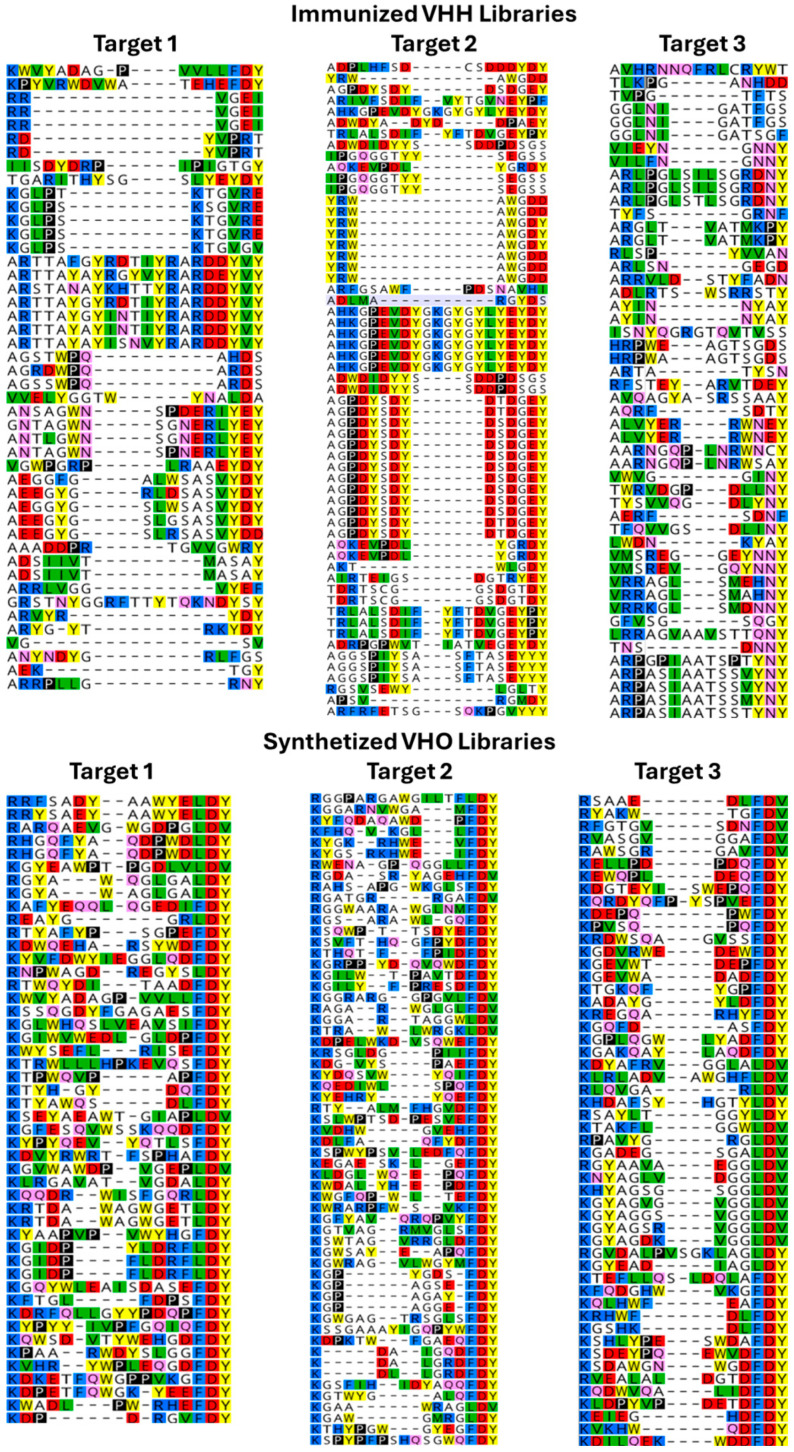
CDR3 diversity comparison between immunized and synthetic sourced phage display libraries [67]. Colors reflect biophysical activity relative to each amino acid: red—negative charge, blue—positive charge, green—hydrophobic, yellow—aromatic, pink—polar uncharged.

**Table 1 antibodies-14-00071-t001:** Profiles of libraries found in the literature.

Library Name	Source of sdAb	Scaffold	Diversified Regions	Library Size	CDR3 Diversity After Biopanning or Screening *** (%)	Number of Bins or Epitopes Per Target
Tornetta et al. [25]	Synthetic	Human VH3	CDR1,2,3	2 × 10^10^	>80 (*n* = 20)	Average of 10
Bahara et al. [26]	Synthetic	Human VH3-23	CDR1,2,3	7 × 10^9^	NA	NA
Pang et al. [27]	Synthetic	Human IGHV 3-23_04	CDR3	2.6 × 10^11^	35–58 (*n =* 2)	At least 4
Sun et al. [28]	Naïve	Human VH3-7, 3-30, 4-34	CDR2,3	1.3 × 10^11^	NA	NA
Bracken et al. [29]	Synthetic	Human 4D5	CDR1,2,3	5 × 10^10^	NA	At least 2
Chen, W. et al. [30]	Synthetic	Human m0_VH3-23	CDR1,2,3	3 × 10^10^	NA	NA
Mindrebo et al. [31] ^+^	Synthetic	Human VH3-23 ^#^	CDR3	3 × 10^9^	NA	At least 4
Rouet et al. [32]	Synthetic	Human VH3	CDR1,2,3	3 × 10^9^	15–90 (*n =* 4)	At least 4
Wu et al. [33]	Synthetic	Human VH3-66.01	CDR1,2,3	NA	NA	At least 3
Zimmerman et al. [34]	Synthetic	camel	CDR1,2,3	9 × 10^12^ *	24–77 (*n =* 3) **	At least 3
Murakami et al. [35]	Synthetic	camel	CDR1,2,3	1 × 10^13^ *	73–81 (*n =* 2)	NA
Sevy et al. [36]	Synthetic	Camel ^˄˄^	CDR1,2,3	1 × 10^9^	12–25 (*n =* 2)	NA
Moutel et al. [37]	Synthetic	Camel ^˄˄^	CDR1,2,3	3 × 10^9^	26–46 (*n =* 2)	NA
Nie et al. [38]	Synthetic	Alpaca ^˄˄^	CDR3	2 × 10^9^	5/17	NA
Contreras et al. [39]	Synthetic	camel	CDR1,2,3	2 × 10^8^	95–100 (*n =* 3)	NA
McMahon et al. [40]	Synthetic	camel	CDR1,2,3	5 × 10^8^	NA	NA
Yan et al. [41]	Synthetic	camel	CDR3	2 × 10^9^	2–7 (*n =* 2)	NA
Chen, X. et al. [42]	Synthetic	camel	CDR1,2,3	>10^10^ *	NA	NA
Arras et al. [43]	Immunization	Llama ^˄˄^	CDR1,2,3	NA	41	NA
Salhi et al. [44]	Immunization	camel	Vh	1 × 10^9^	NA	NA
Ganji et al. [45]	Immunization	camel	Vh	6 × 10^9^	NA	NA
Tsukahara et al. [46]	Immunization	alpaca	Vh	2 × 10^10^	NA	NA
Qiu et al. [47]	Immunization	camel	CDR3	3 × 10^5^	63	NA
Tsukahara et al. [46]	Naïve	alpaca	Vh	2 × 10^10^	NA	NA
Li et al. [48]	Naïve	alpaca	Vh	2 × 10^9^	17	At least 2
Xu et al. [49]	Naïve	alpaca	Vh	3 × 10^9^	37	At least 2

^+^ Yeast display. ^#^ Human framework with some positions changed to camel residues. ^^ Humanized framework. * Ribosome or cDNA display. ** Open reading frame only. *** Unique candidates are defined by having at least 3 different residues across the whole variable region. The “n” represents the number of target biopanning/screening campaigns. Libraries referenced in this table: [25,26,27,28,29,30,31,32,33,34,35,36,37,38,39,40,41,42,43,44,45,46,47,48,49].

**Table 2 antibodies-14-00071-t002:** Sequence assessment of the 2 differently designed libraries.

		Library NGS QC		Selected VHO Uniqueness	
Library Design	Raw MiSeq Reads	Merged 2 × 300 Reads	Correct Merged Sequences *	Searching only with CDR3	Searching the full VHO
1	5,560,747	5,320,435 (95.7%)	3,647,173 (68.6%)	1	0
2	4,882,291	4,654,140 (95.3%)	1,697,155 (36.5%)	25	2
		**Library NGS QC**		**Selected VHO Uniqueness**	
Library Design	Raw NextSeq Reads	Merged 2 × 300 Reads	Correct Merged Sequences *	By Searching only CDR3	Searching the full VHO
1	33,933,864	33,327,254 (98.2%)	21,370,246 (64.1%)	9	2
2	40,256,678	39,223,082 (97.4%)	13,862,286 (35.3%)	19	1

* The correct merged sequences were a compilation of no stop codons, no extra Cysteine, and those containing designed CDR lengths. NGS—next-generation sequence, QC—quality control, VHO—variable heavy only, CDR3—third complementarity determining region.

**Table 3 antibodies-14-00071-t003:** A summary of results after 12 VHO biopanning projects. Each VHO has >4 different residues within CDR3 (Kabat numbering). Project TAA2 performed both protein and cell panning. Not all VHOs were tested against the ortholog proteins (blanks).

Tumor Associated Antigen	VHO	VHO-Fc	Binding to Protein (BLI and/or ELISA)			Epitope/Activitybins
	Candidates	Expressed	Human	Monkey	Mouse/Rat	(at Least)
TAA1	64	65%	90%		90%	7 ^^^
TAA2	64	84%	85%	83%	92%	10
TAA3	48	67%	99%	66%	50%	15
TAA4	80	96%	85%		83%	10
TAA5	110	96%	99%	99%	99%	12
TAA6	110	99%	97%	98%	65%	11
TAA7	128	90%	76%		87%	13
TAA8	64	97%	94%	97%	22%	4 ^^^
TAA9	64	92%	97%		89%	9 ^^^
TAA10	64	81%	75%	35%	89%	10 ^^^
TAA11	64	72%	85%		78%	8 ^^^
TAA12	54	87%	96%			10 ^^^

^^^ Final bin number determined from cell binding results. BLI—biolayer interferometry.

## Data Availability

The data that support the findings of this study are available on request from the corresponding author.

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
