# Peer review of "Guidelines in the Preparation of Fully Synthetic, Human Single-Domain Antibody Phage Display Libraries"

_2073-4468, 2025, doi:10.3390/antib14030071_

Round 1
Reviewer 1 Report (Previous Reviewer 1)
Comments and Suggestions for Authors
This is a revised version of a manuscript that I previously rejected. The authors describe in detail synthesis, quality control, and developability issues of nanobody libraries based on a human framework. This is a very methodological paper but results may be of value for researchers in the field of synthetic antibody library generation.
Importantly, the authors addressed all of my many critical statements in the first version of the manuscript. I can now recommend publication.
(funny) TYPO to be corrected: Li 89: perfromance
Author Response
Thank you for your review of our manuscript and for the reply.
Reviewer 2 Report (New Reviewer)
Comments and Suggestions for Authors
The manuscript titled “Experience In The Preparation Of Efficient And Effective Fully Synthetic Human Single Domain Antibody Phage Display Libraries” by Tornetta et al describes the construction, designing and validation of fully synthetic human single-domain antibody (sdAb) libraries using a human IGHV3-based approach. This work has a high practical relevance as preparation of fully synthetic sdAb has always been a requirement for therapeutic antibody discovery. However, I think some minor structural refinement and improved clarity will help to enhance quality of the manuscript even more. Below I have mentioned areas of improvement of this manuscript:
(1) The title of the manuscript starts with the word “experience” which is a bit of unconventional. The authors are suggested to replace the word with “construction “or “preparation” or something like that to make it sound more direct and robust.
(2) Due to extensive methodological explanation provided in the manuscript, the core findings sometimes seem obscured. The authors should summarize the key findings earlier in the ‘Abstract’ and in the ‘Results’.
(3) The authors are also requested to shorten the overly long sentences. Sometimes, the sentences are sounding conversational. The formal academic tone should be maintained throughout the manuscript. The paragraph transitions can also be improved.
Addressing these following concerns will make this manuscript suitable for publication in the journal.
Comments on the Quality of English LanguageThe authors are also requested to shorten the overly long sentences. Sometimes, the sentences are sounding conversational. The formal academic tone should be maintained throughout the manuscript. The paragraph transitions can also be improved.
Author Response
July 22, 2025
Dear Mr. Wasin Pattaraprachyakul, Section Managing Editor
We want to thank the reviewers for their input that has improved the quality of this manuscript. Most of the revisions addressed the style of writing.
- The title was changed as requested.
- “Core findings” are better organized within the rewritten Abstract which leads the reader into the manuscript to find more details in the Results and Discussion sections.
- We placed the summary of the “core findings” in the beginning of the Discussion section instead of the Results section.
- Many ‘conversational’ sentences are now more direct.
- Paragraphs have been rearranged and rewritten for smoother transitions.
This latest revised version of the manuscript is edited with Tracked Changes.
Sincerely,
Mark Tornetta

This manuscript is a resubmission of an earlier submission. The following is a list of the peer review reports and author responses from that submission.
Round 1
Reviewer 1 Report
Comments and Suggestions for Authors
The authors compiled in retrospect their experimental experience with the use of two hin house libraries with randomized CDRs based on a human scaffold to generate humanized single domain antibody libraries that were used to screen for binders against numerous targets. The libraries were characterized with respect to sequence diversity and screening performance using phage display. I have several severe problems with that manuscript.
- I cannot see sufficient scientific novelty for the general use of phage display libraries that have not been addressed before and therefore would merit publication.
- Since the libraries will not be freely available, the in depth analysis of their composition and performance is only of value for the company (and perhaps their collaboration partners) but not for a broad audience of readers.
- The hits they obtained by library screening are not characterized with respect to affinity and specificity (off-target binding).
- The introduction section, particularly chapter 1.1. has to be entirely rewritten. I find it very problematic since it lacks clarity at many points and I also disagree with several statements made there:
P2 li 67 Unfortunately, more current data is showing that these highly potent anti bodies also interact to on-target, off-disease cells/tissues… This statement reads as if the potential problem of off-target effects would be related to antibodies obtained from immunized animal libraries. But this holds for any antibody irrespective of the source since it is a target problem rather than an antibody problem. The sentence should be clarified or deleted.
P2 Li 70 Besides the understood outcome of low diversity and inconsistent candidate production from most immunizations … I fully disagree. There may be low diversity when you count different sequences by NGS but the hit rate is enormously high. In most cases particularly with VHH there is a broad epitope coverage. The problem only shows up when applying hybridoma technology for antibody generation from immunized rodents but even here broad diversity and epitope coverage can be achieved using direct B cell cloning e.g. by using the Beacon device. What may become a true disadvantage of animal immunization that is not mentioned directly is the problem of self- and nonself discrimination. In case that the animal produces a protein itself that has a sequence close to the antigen used for immunization, B cells targeting identical epitopes may be eliminated. In this case it may be an advantage to use animal species for immunization that are evolutionary distant to humans such as chicken or camelids.
P2 Li 71: there are additional challenges with the downstream processes to capture antibodies… What should these be? For almost all antibody formats (including VHH there exist purification options (Protein A, Protein L. Protein G, Capture Select).
P2 Li 74: On top of this, especially when working with 2 domain antibodies, there is the challenge of pairing the heavy and light chains… it should be mentioned that this problem can be overcome by B cell cloning (Beacon and other systems).
P2 Li 77: It is a strong assumption that yeast or mammalian display will not accept all antibody frameworks since we observe inconsistent expression in the later stages of R&D… I disagree. Indeed some antibodies can be displayed but cannot be expressed. However, this is a rare event that cannot be directly related to particular antibody framework families but can also be influenced by the CDR length, sequence and conformation.
P2 Li 83: The synthetic design approach creates extremely high diversities. This overcomes the bias of the animal’s response or lack thereof to an antigen…. This sounds like an advantage at first glance but an extremely high diversity also results in an extremely low hit rate compared to animal immunization requiring the generation and screening of very large libraries.
P2 Li 87: Cross-reactivity is rarely obtained from immunized and naive sourced libraries. I do not fully agree. It is a standard procedure to immunize with a human antigen and to do a boost immunization with e.g. the cyno antigen to obtain cross-reactive antibodies.
One major disadvantage of animal immunization compared to synthetic libraries that is not mentioned I this section of the paper is the requirement of humanization of the obtained antibodies which is of course not required when using synthetic human-derived libraries.
P3 Li 97: The comprehensive use of codons plays a big part in enabling both stability… How should codon usage influence antibody stability? The encoded amino acids are the same. Or is mRNA stability meant? Then this should be explicitly mentioned.
P3 Li 105: Once a “behaved” framework(s) is established for expression and functional activity, it is then ready to receive a designed CDR library. This statement relates obviously to synthetic libraries. It should be mentioned in the manuscript that synthetic libraries are discussed from this point on
P3 Li 107: There are those that follow the lates mutational patterns found in nature (Distributed Bio), restricted ones for better … Citations are missing.
P3 Li 110: ….is introducing gradual diversity from each CDR, which ultimately leads to the most diverse CDR3 region... Please define and explain gradual diversity.
P3 Li 118: Another question to keep in mind when looking at library design, is there a trend between CDR diversity and/or how that diversity is distributed within the CDRs to a particular germline framework(s). Not understandable. Please explain.
In general, it would interesting to mention how many approved antibodies are derived from animal immunization and how many from synthetic libraries (single digit number).
P3 Li 127: Pairing up VH and VL domains has been an ongoing challenge… Imprecise wording. The problem is not the pairing but the random generation of light chain heavy chain pairs (e.g. when using classical phage display antibody library generation) resulting in huge combinatorial clone numbers of which only a tiny fraction encompasses the the original pairing in B cells.
P3 Li 127: Both chains have shown to be more utilized than others from natural sources (MorphoSys and Distributed Bio)…. I do not understand this statement. Which chains are authors referring to?
P3 Li 139: It is hard to find synthetic sdAb libraries that are being produced and even more so, any that are currently in use. This statement is wrong. See synthetic sdAb libraries generated by the Seeger lab in Zurich (and others).
P4 Li 144: Obtaining enough of an encoded DNA repertoire from an animal source to build a highly diverse phage display library long term is not easy. Disagree, it is easy to transfer the antibody diversity from an immunized camelid to an phage or yeast display format. This is due to the fact that compared to a synthetic library significantly less clones are required for full antibody diversity coverage.
P 4 Li 146 and whole paragraph: Most likely the biggest challenge lies with germline and phage display compatibility…. I do not see any challenge. Obtaining libraries with broad coverage is for most campaigns a trivial task (see the myriads of papers from Adimab and Sanofi). An inherent challenge may be finding an immunization strategy that results in a good immune response. But also for this nowadays many immunization strategies are available up to whole cell or RNA and DNA immunization for obtaining antibodies against difficult targets (membrane proteins).
P4 Li 155: and they need to be adjusted at the right time… Unclear statement. What is meant with “at the right time?”
P4 Li 155-158 Rephrasing what has been stated before.
Reference to Figure 1 is missing in the text.
Table 1: In the column library name the number of the cited paper should be given.
P9 li 372: typo diversity of 105 li 374 1011 to 1019 nothing uppercase (also in following sections). -80oC
- NGS Sequencing: Illumina technology was used. This normally generates rather short reads and I wonder whether the reads are sufficiently long to allow for the full assembly of CDR1, CDR2, and CDR3 of different sdAbs
- Only at p 10 line 401 it becomes clear that library 1 was generated via trincleotide codon randomization in contrast to library 2. No citation and material and methods information is provided for Design 1.
Comments on the Quality of English Language
English writing is acceptable but could be improved.
Reviewer 2 Report
Comments and Suggestions for Authors
Introduction
Compared to original research articles in the general life sciences field, the Introduction in this paper is considerably long. While it is commendable that the methodology and background knowledge are comprehensively and in detail explained, this paper is an Article, not a Review. Instead of providing an overview of the entire research field, the focus should be narrowed down to literature and findings that are directly relevant to the main subject of this study.
Ideally, the latter part of the Introduction should clearly state "what question this research aims to address and how it intends to solve or verify it," and explain the novelty and objectives of the study. The absence of such a statement is critical. The Introduction merely ends with background explanations such as “synthetic libraries have advantages” and “immunized libraries have these disadvantages,” which results in an ambiguous final research objective. For readers, despite the extensive content, it is unclear what is new in this study and how it advances previous work.
In particular, Section 1.2 discusses important parameters for library construction, but their necessity and significance are not clear. For example, while there is no disagreement on the importance of service providers per se, I do not think it is necessary to describe that in the Introduction of this study. Some parts of the content could serve as comparative material in the Discussion to support claims regarding the validity or superiority of the parameters adopted in this study.
On the other hand, while the Introduction spans seven pages, the Discussion is only two and a half pages long, which is extremely unbalanced. In my opinion, several of the points described in the Introduction should be moved to the Discussion for further debate. Much of the content should be utilized to compare existing research or to discuss limitations.
In conclusion, for this paper to be accepted as an Article, it is necessary to selectively condense and significantly shorten the information currently included in the Introduction. For most readers, an Article’s Introduction should concisely and clearly summarize the key points—what the challenges are, why this study is needed, and what aspects are novel—rather than being detailed and lengthy. Detailed technical explanations and historical background on methodologies should be distributed into the Methods or Discussion sections, or moved to the Supplementary Information, so that the Introduction can be kept concise with sections on “research background,” “problem statement,” “objectives/hypotheses,” and “strategic overview.”
Line 128, 129:
The source is cited by listing company names, but specific papers should be cited instead.
Line 67:
The authors state that antibodies obtained from animal sourced libraries exhibit "low efficacy behaviors" due to off-target effects. However, it is highly questionable whether such a consensus is widely accepted among antibody researchers to warrant inclusion in the Introduction. In my opinion, these issues are more likely the result of improper enrichment or antibody selection from the libraries. Even if off-target effects are observed for a particular antibody selected from an animal sourced library, it is rather rash to conclude that the majority of antibodies in that same library exhibit the same trend. Moreover, is there sufficient data to statistically prove that similar tendencies are generally present in other animal sourced libraries, with significantly more non-specific binding compared to semi-synthetic or synthetic libraries? Unfortunately, the paper cited by the authors as evidence is a paper on the construction and performance evaluation of a synthetic VHH library, and it does not provide direct experimental data on the phenomenon of "on-target, off-disease" interactions in antibodies derived from animal immunization. If the authors wish to argue that synthetic libraries are particularly suited to obtaining antibodies with extremely high specificity, this should be discussed in the Discussion section.
Line 166 and Table 1:
I do not feel it is necessary to cover existing examples of libraries in such detail in the Introduction. I recommend moving this content to the Discussion. The table should also include the results from this study for a comparative discussion with past libraries, addressing the usefulness of the technique and the evaluation criteria.
Table 1:
For the "Library name" column, the authors of the published papers are listed, but the corresponding reference information is not provided. Please add citation details to Table 1.
Results
Not only are objective results presented, but the authors’ subjective opinions and claims are also interspersed throughout the section. In parts of "3.3 Biopanning Output" and "3.4 Mammalian Expression," passages that discuss the methodological concepts, as well as the interpretation and significance of the results, contaminate the section. The Results section should simply present the observed facts concisely, while the interpretation and implications should be reserved for the Discussion. Additionally, content that overlaps with the Introduction or belongs in the Methods section is also mixed in.
There is insufficient presentation of specific results—such as quantitative data or figures that support the authors’ claims—which weakens the reliability and persuasiveness of the findings. For example, showing the enrichment profile of biopanning, the chromatograms from SEC, SDS-PAGE images of produced VHO-Fc, sensorgrams from BLI, or a distribution of Kd values would greatly enhance the impact. It would also be beneficial to present a graph of the average hydrophobicity of the entire variable region mentioned in the text.
Redundant content and sentences are found across the subsections (3.1 Library Design, 3.2 Library QC, 3.3 Biopanning Output, 3.4 Mammalian Expression, 3.5 Candidate Activity Results).
Overall, while the paper makes an impact by presenting large numerical values, it lacks sufficient concrete examples and quantitative details. I recommend considering a case study approach for one or two TAA targets to present more specific and detailed results—such as the progression of enrichment during biopanning, the sequences and characteristics of clones that notably increased in NGS, as well as the final affinity and expression levels.
Discussion
There is a lack of interpretation, implications, and comparisons with other studies; much of the discussion merely reiterates the content from the Results section. The focus is on listing data without adequately developing the discussion, so the claim that the synthetic library is superior is not convincingly supported. Even if the results are reiterated, they should be summarized concisely as facts, and then the mechanisms behind these outcomes and differentiation from previous studies, including a logical, thorough discussion, should be provided. Furthermore, comparisons with prior research are minimal.
The discussion should employ appropriate citations to deepen the comparison with other methods and clarify the positioning and differentiation of the authors' data.
Although there is a paragraph addressing limitations and constraints, it is insufficiently detailed.
Line 557:
The text states that the low affinity and hit rate of the synthetic library have been overcome by machine learning, as if that were an established fact; however, no supporting data or citations are provided in the paper. This claim may deviate from the content of the current study. If machine learning is indeed employed in this research, please provide details of the methods and results—such as the input data, learning objectives, model performance metrics, and the actual outcomes of the sequence selection. If machine learning is not currently used, it should be clearly positioned as a future prospect to avoid confusing the reader.
Typographical Errors
This manuscript contains numerous typographical errors. Below are just a few examples that I have noticed. In addition to the authors’ own review, professional third-party proofreading should be obtained to eliminate these errors and ensure that the manuscript is presented in a format appropriate for a scientific publication.
The notation for exponents is not correctly formatted; the superscript is not used (e.g., in Lines 374, 376, 390, and many others).
There are unnatural spaces, for example, in Lines 128 and 129.
In Line 395, the degree symbol is not formatted as a superscript.
In Table 1, the exponent notation for library sizes is indicated using the '^' symbol.
Figures
Figure 1
The resolution is low, and the text is particularly difficult to read. The caption states, “Figure 1. Types of sourced antibody phage libraries [15],” but it is unclear what exactly from reference [15] is being cited. I assumed that the figure was taken directly from the cited paper; however, there is no identical figure in reference [15].
Figure 3
The figure is too large. Both the sequence logo and the graph can be made smaller while remaining legible. It is preferable that the figure and its legend are contained on the same page. The graph in Panel B appears to use Excel’s default formatting, giving the impression that it was cut and pasted directly from an experimental notebook. Please add the appropriate details (such as axis titles with units, display the vertical axis, add tick marks, etc.) and improve the formatting. Also, clarify whether “#VHOs” means “Number of VHOs.”
Figure 4
The text is nearly unreadable due to the low resolution and excessive noise. The figure should be re-edited to achieve an appropriate resolution and size.